# Late-Onset Medullary Thyroid Cancer in a Patient with a Germline *RET* Codon C634R Mutation

**DOI:** 10.3390/diagnostics11081448

**Published:** 2021-08-11

**Authors:** Agnieszka Walczyk, Kajetan Zgubieński, Grzegorz Chmielewski, Kinga Hińcza-Nowak, Artur Kowalik, Jarosław Jaskulski, Aldona Kowalska

**Affiliations:** 1Endocrinology Clinic, Holycross Cancer Center, S. Artwińskiego St. 3, 25-734 Kielce, Poland; aldonako@onkol.kielce.pl; 2Collegium Medicum, Jan Kochanowski University, IX Wieków Kielc Av. 19, 25-319 Kielce, Poland; kzgubieski@gmail.com (K.Z.); grzegorz.chmielewski1@gmail.com (G.C.); jaroslawja@poczta.fm (J.J.); 3Department of Molecular Diagnostics, Holycross Cancer Center, S. Artwińskiego St. 3, 25-734 Kielce, Poland; kinga.hin@wp.pl (K.H.-N.); Artur.Kowalik@onkol.kielce.pl (A.K.); 4Division of Medical Biology, Institute of Biology, Jan Kochanowski University, Uniwersytecka St. 7, 25-406 Kielce, Poland

**Keywords:** multiple endocrine neoplasia type 2A, hereditary medullary thyroid cancer, germline C634R *RET* mutation, genotype-phenotype correlation, risk stratification

## Abstract

**Background**: Multiple endocrine neoplasia type 2A (MEN2A) is a rare, hereditary syndrome resulting from a germline mutation in the *RET* proto-oncogene and characterized primarily by medullary thyroid cancer (MTC), pheochromocytoma (PHEO), and hyperparathyroidism. Types of *RET* mutation have been associated with age at onset, clinical outcomes of MTC, and the penetrance of other components. Patients classified as ‘high-risk’ by the American Thyroid Association (ATA), based on the aggressiveness of MTC and the penetrance of other components, are recommended to undergo early prophylactic thyroidectomy at age ≤ 5 years and to be screened for PHEO at age ≥ 11 years. Patients with *RET* codon C634R mutations have been classified as high-risk. **Case presentation**: The present study describes a 71-year-old woman newly diagnosed with hereditary MTC related to a *RET* C634R germline mutation. Her basal serum calcitonin level was high, but there was no evidence of distant metastases. Surgery revealed bilateral MTC with two metastatic lymph nodes. Because microscopic resection was incomplete and extranodal extension was observed, the patient underwent adjuvant external beam radiotherapy. Response to therapy was excellent. Follow-up after 1.5 years showed no evidence of disease or other manifestations of MEN2A. **Conclusion**: Despite *RET* C634R carriers being classified as high-risk by the ATA, this patient did not present with either distant MTC or PHEO until her seventies. To our knowledge, only one other patient has shown a similar late identification of a *RET* C634R mutation, but MTC could not be diagnosed because the patient was lost to follow-up. Further research is required to develop optimal protocols that could allow patients requiring prophylactic thyroidectomy to be differentiated from those who can be monitored closely without early surgery.

## 1. Introduction

Multiple endocrine neoplasia type 2A (MEN2A) is a rare, autosomal dominant hereditary syndrome associated with tumors of the endocrine glands, the most typical being medullary thyroid cancer (MTC), pheochromocytoma (PHEO), and hyperparathyroidism (HPTH). In Europe, the prevalence of MEN2A ranges from approximately 13 to 24 per 1,000,000 people [1,2]. Other disorders, such as cutaneous lichen amyloidosis and Hirschsprung’s disease, may also occur in patients with MEN2A [3]. The diagnosis of MEN2A is based on clinical features and genetic testing. Endocrine disorders are caused by various germline mutations in the *RET* (Re-arranged during T-ransfection) proto-oncogene [4,5]. Types of *RET* mutations result in various disease phenotypes, with differences in age at onset, clinical presentation (including the aggressiveness of MTC), and outcomes of patients with MEN2A. MTC is the major prognostic factor, as well as the leading cause of death, in patients with MEN2A, despite the fact that genetic testing at an early age and appropriate prophylactic thyroid surgery have reduced death rates in patients with hereditary MTC from 15–20% to 5% [6,7]. The American Thyroid Association (ATA) has classified hereditary MTC into three risk categories: moderate (ATA-MOD), high risk (ATA-H), and highest risk (ATA-HST) [3]. Risk category has been associated with the location of the *RET* mutation, including the exon or codon mutated as well as the type of amino acid substitution. In particular, any *RET* mutation at codon 634, located in exon 11, and the A883F mutation (replacement of alanine by phenylalanine) at codon 833 of exon 15, were found to correlate with significant risk and have been classified as high risk. Furthermore, specific amino acid substitutions at codon 634 have been associated with more aggressive phenotypes of MTC. For example, the replacement of cysteine by arginine at *RET* codon 634 (C634R) was found to show higher association with the occurrence of distant metastases of MTC at presentation than the replacement of cysteine by tyrosine (C634Y) or tryptophan (C634W) [8,9]. MTC is usually diagnosed in carriers of C634R *RET* mutations during the second or the third decade of life [10]. However, individuals with C634R mutations present with metastatic lymph nodes and distant metastases significantly earlier than those with C634Y/W mutations [8]. The ATA therefore recommends that children classified as ATA-H undergo genetic testing and, if warranted, prophylactic thyroidectomy at age ≤ 5 years, with the timing and extent of thyroid surgery based on serum calcitonin (Ctn) concentrations [3]. Mutations at *RET* codon 634 are also associated with a high penetrance of PHEO, with positive age-related rates of PHEO at ages 30, 50, and 77 years being 25%, 50%, and 88%, respectively [11]. Prior to genetic testing for MEN2, the risk of death associated with an undiagnosed PHEO in unidentified *RET* carriers was estimated as high as the risk of death from progressive MTC [7,12]. This report describes a 71-year-old woman with a late diagnosis of regionally advanced hereditary MTC related to a germline *RET* codon C634R mutation but without a confirmed diagnosis of PHEO or HPHT.

## 2. Case Presentation

A 71-year-old woman was referred to the Endocrinology Clinic of Holycross Cancer Center in Kielce, Poland, in October 2019 by her granddaughter, who was a carrier of the germline C634R *RET* mutation and was newly diagnosed with MEN2A syndrome. The patient’s family history of the germline C634R *RET* mutation was well determined, because the patient’s three siblings and their numerous children and grandchildren were positive for this *RET* mutation, but the present patient was not referred to our Center until now. Her medical history included a diagnosis of type 2 diabetes and dizziness, but she was negative for hypertension, palpitation, and headache. A physically palpable painless mass was present in her right thyroid lobe, whereas her blood pressure (BP: 140/80 mmHg) and heart rate (84 beats per minute) were normal. Sanger sequencing of her genomic DNA showed a germline mutation, p.(Cys634Arg) c.1900T>C, in exon 11 of the *RET* proto-oncogene. (Figure 1). Genomic DNA was isolated from whole peripheral blood using Maxwell RSC Blood DNA Kits (Promega, Madison, WI, USA), according to the manufacturer’s instructions. DNA quality and purity were verified using a NanoDrop spectrophotometer (Thermo Fisher Scientific, Waltham, MA, USA).

*RET* exon 11 was PCR-amplified using the PCR primers 5′-CCT CTG GCGG TGC CAA GCC TC-3′ (forward) and 5′-GAC CCT CAC CAG GAT CTT GAA GGC-3′ (reverse). The amplification conditions consisted of an initial denaturation at 95 °C for 3 min, 35 cycles of denaturation at 95 °C for 30 s, annealing at 68 °C for 30 s, and extension at 72 °C for 30 s, and a final extension at 72 °C for 3 min, performed using a Veriti Thermal Cycler (Thermo Fisher Scientific). The PCR products were separated by microchip electrophoresis, using a MultiNA System (Shimadzu Corporation, SHIM-POL, Izabelin, Warsaw, Poland) and purified using FastAP and exonuclease I (Thermo Fisher Scientific). The purified PCR products were sequenced using a BigDye Terminator v1.1 Cycle Sequencing kit (Thermo Fisher Scientific) and the same forward and reverse primers, only the primers were diluted in a ratio of 8:42 μL in water. The PCR products were subsequently separated and analyzed using a 3500 Series Genetic Analyzer (Thermo Fisher Scientific).

Laboratory examination before surgery revealed euthyroidism, high concentrations of Ctn and carcinoembryonic antigen (CEA), slightly elevated concentrations of parathyroid hormone and metanephrine in only one 24 h urinary test, and a mild vitamin D3 deficiency (Table 1).

Because PHEO was suspected, the 24 h urinary test was repeated, showing normal concentrations of metanephrine. Ultrasound (US) of the thyroid gland revealed two solid hypoechogenic lesions in the upper part of the right lobe, the first one measuring 35 × 29 × 41 mm with macrocalcifications, and the second measuring 22 mm in maximum diameter without calcifications. US also showed a hypoechogenic lesion, measuring 9 × 7 × 9 mm, in the left lobe (Figure 2), but no lymph nodes suggesting regional metastases. A risk of malignancy of each thyroid nodules was estimated according to the EU-TIRADS classification system [13]. A chest X-ray was normal, whereas abdominal computed tomography (CT) revealed a focal lesion, measuring 12 × 9 × 13 mm, in the left adrenal gland, with non-contrast and contrast-enhanced attenuation levels of 24 and 78 Hounsfield units, respectively (Figure 3).

Because CT imaging did not result in a clear exclusion of PHEO, magnetic resonance imaging (MRI) with contrast enhancement was performed, showing segmental thickening of the medial branch of the left adrenal gland and a solid focal lesion, measuring 11 × 9 mm and with well-defined edges, indicative of a low-fat adenoma without any characteristics of PHEO, so additional procedure such as scintigraphy with metaiodobenzylguanidine (MIBG) was not performed. MRI did not reveal any other significant lesions in the abdomen, including the liver. Fine-needle aspiration (FNA) biopsies of the larger nodule of the right thyroid lobe (Figure 2A) and the focus of the left thyroid lobe (Figure 2C) revealed bilateral MTC (Figure 4A). To confirm the diagnosis of MTC, Ctn concentrations were measured in both FNA washout fluids, but Ctn was significantly higher than normal only in the specimen retrieved from the right lobe (15,417.00 pg/mL).

In March 2020, after clinical exclusion of PHEO, the patient was administered pre-operative alpha-blockade, followed by total thyroidectomy with bilateral neck dissection. Histologically, bilateral MTC was diagnosed, with the lesions in the right and left lobes measuring 40 mm and 19 mm, respectively. Histologically, bilateral MTC of stage pT2N1b (Figure 4C–E) with angioinvasion and positive margin status (R1) was recognized. Two metastatic lymph nodes, measuring 4 mm in maximum diameter, were detected in the right cervical lateral compartment, accompanied by extranodal extension, but no metastatic nodes were detected in the central and left cervical compartments. Multifocal C-cell hyperplasia was also identified in the left thyroid lobe and isthmus (Figure 4B).

A postoperative evaluation in May 2020 showed that basal Ctn and CEA concentrations were slightly elevated, whereas the concentration of metanephrine in a 24 h urinary test was normal. Because surgery was incomplete microscopically and extranodal extension was detected, the patient was referred for adjuvant external beam radiation therapy, which consisted of a total dose of 66 Gy in 30 fractions and was well tolerated. Follow-up in August 2020 showed that both Ctn (4.63 pg/mL, normal 0–5.0 pg/mL) and CEA (1.43 ng/mL, normal 0–5.0 ng/mL) levels were within normal ranges, and there was no structural evidence of disease. At the time of writing, the patient remains in remission of MTC.

## 3. Discussion

*RET* proto-oncogene germline mutations were first thought to be involved in the development of hereditary MTC in 1990, with their involvement confirmed in 1993 [4,14,15]. The Gubbio consensus in 2001 recommended that the timing of prophylactic thyroid surgery be correlated with the type of germline *RET* mutation [7]. Hereditary MTCs were classified into three risk categories, with risk and the optimal timing of surgery being associated with the type and location of specific *RET* mutations. Patients with mutations in codon 634 of *RET* have been classified as high-risk for aggressive MTC, with children carrying the mutation recommended to undergo thyroidectomy before the age of 5 years [7]. Further analysis in 2006 suggested raising the risk levels for C634R mutations and performing thyroidectomy at younger ages [16].

In 2009, the ATA classified patients into four risk levels, A, B, C and D, based on specific *RET* mutations [17]. Patients with MTC classified into levels A and B, and normal Ctn concentration, normal neck US, and a negative family history of aggressive MTC were allowed to delay surgery until after the age of 5 years. Levels C and D were associated with higher risk of development of aggressive MTC, with level C including only patients with *RET* codon 634 mutations. Surgery at ages <5 and <1 years have been recommended for patients classified as levels C and D, respectively. Although the present patient was classified as level C, she did not require prophylactic thyroidectomy, providing further evidence that patients carrying the same mutation may differ in progression to clinically overt MTC, even within the same family [18].

To enhance current specific *RET*-related management approaches, the ATA recommended new risk categories for hereditary MTC in 2015 [3]. These categories recommended that the timing and extent of prophylactic surgery depend not only on *RET* genotype, but also on Ctn levels, physical examination, neck US, and a clinical presentation of hereditary MTC in particular families. The previous level C has been changed to a high-risk category, called ATA-H. In addition, patients with the *RET* codon A883F mutation, who had been classified as the highest level, D, were re-classified as ATA-H. However, the most recent ATA guidelines, along with European recommendations on the timing of thyroidectomy in high-risk patients, have been maintained. Based on basal serum Ctn concentrations, prophylactic thyroidectomy has been recommended at age ≤5 years for children classified as ATA-H [3,19].

Ctn is a sensitive and specific marker of MTC that plays a crucial role in diagnosis and therapeutic management, with Ctn concentration strictly correlating with tumor mass [20,21]. These findings have suggested that the timing of prophylactic thyroidectomy be based on preoperative basal Ctn concentrations rather than on risk category, except for patients in the highest risk category [22]. Nevertheless, the current recommendations suggest that the need for prophylactic thyroidectomy in patients with MEN2A, differing slightly in optimal timing, be based on genotype-phenotype correlations, although one study found that only *RET* codon 634 mutations were significantly associated with a high risk of MTC recurrence, in spite of classifying other mutations in *RET* codons 611, 618, and 620 as being at the same risk level and requiring early thyroid surgery [23]. Tumor aggressiveness and patient outcomes have been found to depend on specific amino acid substitutions at codon 634, with the C634R mutation being associated with a more aggressive phenotype, manifesting as a higher penetrance of MTC, PHEO, and HPHT at a younger age [9]. Furthermore, carriers of *RET* codon 634 mutations were found to be at the highest risk of development of HPHT, up to 20–30% [3,24]. To date, neither PHEO nor HPHT has been diagnosed in the present patient, aged 72 years now.

The patient described in this report represents a very late onset of MTC in a germline carrier of a *RET* codon C634R mutation. This patient, first diagnosed at age 71 years, was unusual, for both the late age of onset and no need for prophylactic thyroidectomy to improve prognosis and overall survival. Until the age of 71 years, this patient did not require medical care for any condition related to this germline *RET* mutation. Moreover, she underwent treatment without any severe adverse event. Despite her age, the pathological stage of MTC (pT2N1b) and clinical M0 made possible a successful clinical outcome of surgery followed by adjuvant radiotherapy. Furthermore, follow-up assessment based on European Society for Medical Oncology guidelines showed that her response to treatment was excellent, with Ctn and CEA within reference ranges 6 months after surgery, and US showing no structural evidence of disease [25]. The omission of prophylactic thyroidectomy in this patient likely did not shorten her lifespan or worsen her quality of life. Rather, it prevented years of thyroid replacement therapy and potential complications related to early thyroidectomy, with the risk of chronic postoperative hypoparathyroidism being the most important, especially when surgery is extensive and includes lymph node dissection [26]. In addition, this patient did not develop any other characteristic manifestation of MEN2A, such as PHEO or HPTH, which usually appear in young adult carriers of germline *RET* C634 mutations.

To our knowledge, there has been only one previous case report of a late diagnosis of heterozygous polymorphism at codon C634R on exon 11 rs377767437 sequence of the *RET* proto-oncogene. That patient, aged 73 years, presented with HPTH, bilateral PHEO, and a thyroid nodule. Unfortunately, that patient refused treatment for the thyroid lesion and was lost to follow-up [27].

## 4. Conclusions

Single case reports are far from enough to challenge well-established and documented guidelines. Nevertheless, research is required to determine whether all patients with mutations perceived as ‘aggressive’ eventually develop MTC. Although the ATA and other guidelines have stated that the development of MTC is virtually certain in ATA-H MEN2A patients, findings in the present and other patients indicate that the possibility of a late onset of MEN2A components cannot be ruled out. Further research is required to develop optimal protocols that could allow patients requiring prophylactic thyroidectomy to be differentiated from those who can be monitored closely without early surgery. The latter may be an option for children whose parents do not consent to prophylactic thyroidectomy at the recommended early age.

## Figures and Tables

**Figure 1 diagnostics-11-01448-f001:**
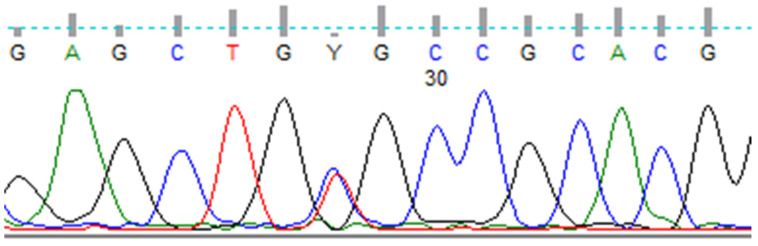
Sanger sequencing of codon 634 of *RET* showing p.(Cys634Arg) c.1900T>C mutation.

**Figure 2 diagnostics-11-01448-f002:**
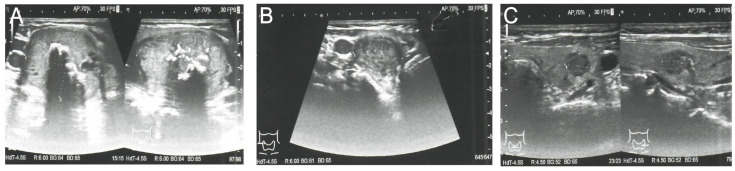
Preoperative ultrasound of the thyroid gland and the EU-TIRADS categories of thyroid nodules. (**A**) Right lobe, showing the dominant thyroid nodule, measuring 35 × 29 × 41 mm, with macrocalcifications; EU-TIRADS 4. (**B**) Right lobe, showing a second hypoechogenic nodule 22 mm in width and 16 mm in height; EU-TIRADS 4. (**C**) Left lobe, showing a hypoechogenic nodule, measuring 9 × 7 × 9 mm, with irregular margins; EU-TIRADS 5.

**Figure 3 diagnostics-11-01448-f003:**
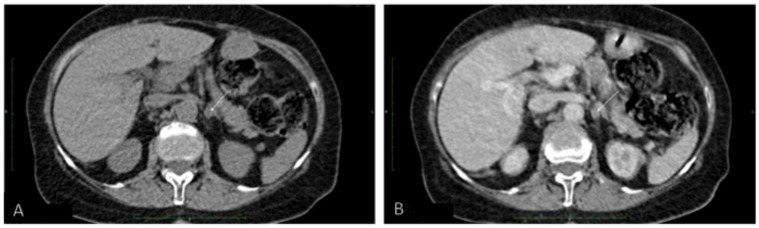
Abdominal computed tomography (**A**) without and (**B**) with contrast of a focal lesion in the left adrenal gland.

**Figure 4 diagnostics-11-01448-f004:**
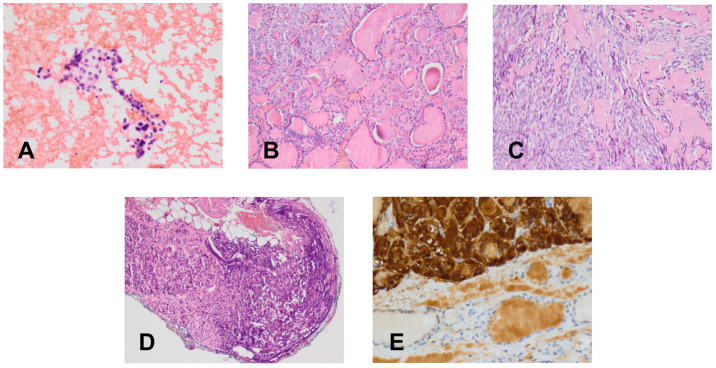
Cytological and histological (H & E) findings (×400). (**A**) Fine-needle aspirate with medullary thyroid cancer. (**B**) Primary C-cell hyperplasia in the patient’s isthmus. (**C**) Medullary thyroid cancer in a thyroid nodule. (**D**) Metastasis of medullary thyroid cancer in a cervical lymph node. (**E**) Immunohistochemistry for calcitonin in a medullary thyroid cancer.

**Table 1 diagnostics-11-01448-t001:** Laboratory findings before and after thyroidectomy in a patient with a germline C634R *RET* mutation identified at the age of 71 years.

Parametere	Reference Range	Before Surgery	After Surgery
Serum calcitonin (pg/mL)	0.00–5.00	1596.00	5.81
Serum CEA (ng/mL)	0.00–5.00	147.69	6.48
Serum calcium (mmol/L)	2.10–2.60	2.46	2.32
Serum phosphorus (mg/dL)	2.70–4.50	3.11	3.22
Serum intact PTH (ng/mL)	15.00–68.30	103.8	67.9
Serum TSH (µIU/mL)	0.35–4.94	0.6455	6.77
Serum Vitamin D (ng/mL)	7.00–53.20	27.70	60.60
Serum CgA (ng/mL)	19.40–98.10	N/A	58.70
Urine metanephrine (µg/24 h) test I/test II	74.00–297.00	251.69/304.37	169.59
Urine normetanephrine (µg/24 h) test I/test II	105.00–354.00	321.89/322.74	264.15

Notes and abbreviations: Preoperatively, 24h urinary metanephrines and normetanephrines were tested twice; CEA, carcinoembryonic antigen; CgA, chromogranin A; N/A, not available; PTH, parathyroid hormone; TSH, thyroid-stimulating hormone.

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
