# Peer review of "Late-Onset Medullary Thyroid Cancer in a Patient with a Germline RET Codon C634R Mutation"

_diagnostics, 2021, doi:10.3390/diagnostics11081448_

Round 1

Reviewer 1 Report

This case report describes a very late onset of a medullary thyroid carcinoma related to a RET C634R germline mutation. The care reports is well documented and clearly exposed. 

Minor comments:

  • It is not clear if the second nodule in the right lobe (figure 2B) had a cytology suspicious for malignancy. Please clarify.
  • Please provide the risk for malignancy of each thyroid nodule (TIRADS score, ATA risk or EU-TIRADS) in the figure and/or in the text. 
  • Referring to the selection of patients that could benefit from a less aggressive management (abstract lines 34-35):  ​​what clinical characteristics could help to identify patients with less aggressive disease?

Author Response

Response to the Reviewer 1

On my behalf and in the name of all co-authors, I would like to thank you for your revision of our manuscript resulting in the constructive guidelines and comments, which will let us improve the manuscript to be accepted for the publication in Diagnostics.

Question 1.

The relevant description has been added to the main text (line 145-146). We specified in which thyroid nodules FNAC was performed.

Question 2.

A risk of malignancy of all thyroid nodules was assessed according to the EU-TIRADS classification system. Both in the text (line 125-127) and in the figure (the caption under the Fig.2) the respective description has been added. Due to this fact, a new reference numbered as [13] and relevant to the ETA guidelines and EU-TIRADS risk stratification system has been added. Subsequent references were re-numbered. 

Question 3.

We agree with the Reviewer’s comment that the statement in the end of abstract may appear slightly unclear.

The Authors’ intention was to pay the Readers’ attention to possibility of a late onset of hereditary MTC, even in the ATA-H risk RET carriers. Potentially, such patients could avoid prophylactic thyroidectomy recommended in an early age in spite the fact that it is associated with higher complication rates in children or infants. But, this single case report is far from a definition of clinical identification of these patients, because it needs further research to identify and create an optimal protocol for them. Thus, the last sentence has been rephrased to clarify the conclusion of the current case report (line 34-36).   

Reviewer 2 Report

Well described case report of C634R RET mutation with late onset of MTC.

Author Response

Response to the Reviewer 2

On my behalf and in the name of all co-authors, I would like to thank you for your revision of our manuscript and the appreciation of the publication in Diagnostics.

Reviewer 3 Report

Agnieszka Walczyk et al described Late onset medullary thyroid cancer in a patient with a 2 germline RET codon C634R mutation.

This article indicate that late onset medullary thyroid cancer in MEN2. However, the authors should describe some information.

  1. The author should describe genetic family tree.
  2. A description of whether or not [131I]/[123I] metaiodobenzylguanidine (MIBG) was performed.
  3. The author should describe the surgical details and histopathology of pheochromocytoma.

Author Response

Response to the Reviewer 3.

On my behalf and in the name of all co-authors, I would like to thank you for your revision of our manuscript resulting in the comments, which will let us improve the manuscript to be accepted for the publication in Diagnostics.

Question 1.

We kindly would like to pay the Reviewer’s attention that the current manuscript present only one case report, not family case report. The RET family history of the presented patient is under our separate scientific interest, and it is preparing to an independent Family Case Report, as it involves close to 20 people. Thus, we limited the current patient’s family history description to a very short report of numerous RET positive relatives to pay the Readers’ attention to the fact that the patient delayed her genetic diagnostics despite known and long-lasting RET positive family history. To our opinion, lack of genetic family tree presentation in details in the manuscript does not diminish a value of case report. The entire genetic tree of this family is worth to be discussed as an independent report and, what is more, it will help us avoid a plagiarism in a future.

Question 2.

In the main text we described in details the diagnostic way of excluding PHEO prior to the thyroid surgery resulting in the recognition of low-fat adenoma in the patient’s left adrenal gland. Thus, there was no need to perform 131I-MIBG. However, according to the Reviewer’s comment this statement has been added to the case description (line 143-144).

Question 3.

As we explained above, we could not describe details surgery and histopathology of PHEO in the present case report, because pheochromocytoma in the presented patient was not identified and the patient did not undergo surgery due to it.

However, according to germline RET mutation positivity, the patient’s follow-up must include screening of PHEO in any case of planned surgery, but the only thyroid (not adrenal) surgery was performed in the reported patient as we described in the manuscript.